# A Min-P Blueprint for More Rigorous Science in Empirical Machine Learning Research

## Abstract

In light of a growing crisis of rigor in empirical machine learning research, this paper provides a blueprint for conducting more meticulous science. We present a detailed case study of "Turning Up the Heat: Min-P Sampling for Creative and Coherent LLM Outputs" (Nguyen et al., 2024), a high-visibility ICLR 2025 Oral paper that introduced a new method for sampling from language models called min-p. The original work claimed that min-p sampling achieves superior quality and diversity over established methods. However, our comprehensive re-examination of the original paper's four main lines of evidence demonstrates that its conclusions are invalidated by its own data. Our re-analysis reveals that: (1) The original human evaluations omitted one-third of the collected data, applied statistical tests incorrectly, and inaccurately described qualitative feedback; a correct analysis shows min-p did not outperform baselines. (2) Extensive hyperparameter sweeps on NLP benchmarks show min-p's claimed superiority vanishes when controlling for the volume of hyperparameter tuning. (3) The LLM-as-a-Judge evaluations suffered from methodological ambiguity and appear to have reported results inconsistently, favoring min-p. (4) Claims of widespread community adoption were found to be unsubstantiated and were retracted. From this case study, we derive a blueprint for more rigorous research. Key lessons include the critical need to compare methods fairly by controlling for hyperparameter tuning, to apply statistical tests transparently and correctly (e.g., correcting for multiple comparisons), to practice full data transparency, and to scrutinize qualitative summaries, methodological clarity, and potentially selective reporting. Adhering to these principles is essential for ensuring the validity of scientific claims and fostering genuine progress in the field of machine learning research.

## 1 Introduction

Machine learning research is currently experiencing crises on multiple fronts (Kim et al., 2025; Schaeffer et al., 2025b): The number of submissions to each conference is skyrocketing (Paper Copilot, 2025), reproducibility concerns are giving rise to a standalone machine learning reproducibility conference (Pineau et al., 2017; Sinha et al., 2023), and scandals concerning prominent publications are proliferating, e.g., (Carlini, 2020; Agarwal et al., 2021; Carlini et al., 2021; 2022; Kirsch, 2022; Schaeffer et al., 2022; Orabona, 2023; Schaeffer et al., 2023; Gerstgrasser et al., 2024; Kirsch, 2024; Markov, 2024; Schaeffer et al., 2024; Chandak et al., 2025; Golechha et al., 2025; Ivanova et al., 2025; Kazdan et al., 2025; Maini & Suri, 2025; Zhang et al., 2025; Schaeffer et al., 2025a).

In this work, we push back against these trends by providing a blueprint for performing more meticulous and rigorous science in empirical machine learning research. Our blueprint is based on a detailed case study of a high-visibility publication: "Turning Up the Heat: Min-P Sampling for Creative and Coherent LLM Outputs" (Nguyen et al., 2024). This publication introduced a method for sampling from language models called min-p sampling, claiming min-p produces higher quality and more diverse outputs than existing sampling methods such as top-k (Fan et al., 2018) or top-p (Holtzman et al., 2020) sampling. The paper ranked as the 18th highest-scoring submission to ICLR 2025 and was selected for an Oral presentation. Given the potential impact of improving both quality *and* diversity of language model outputs, we carefully scrutinized the methodologies, data, analyses, code and conclusions presented in support of min-p across the authors' four lines of evidence: (1) human evaluations, (2) natural language processing (NLP) benchmark evaluations, (3)

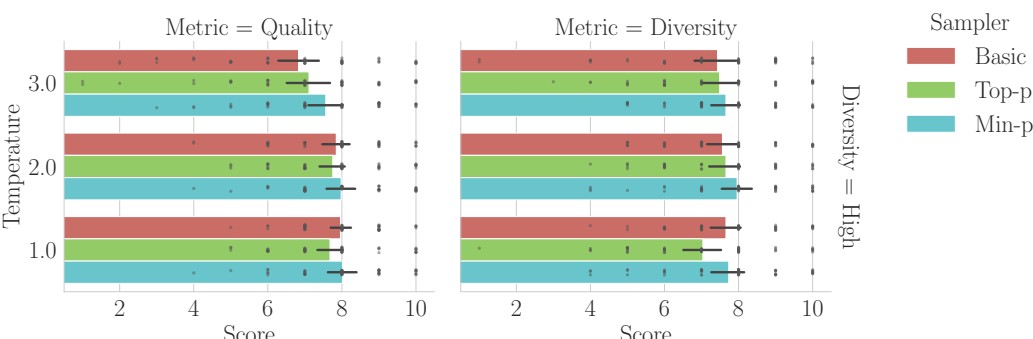

Figure 1: **Visualizing Human Evaluators' Scores from Nguyen et al. (2024)'s Data Demonstrates `Min-p` Does Not "Consistently" Outperform Other Samplers.** Rather, the original paper's data suggest `min-p` is largely indistinguishable from other samplers based on 95% confidence intervals.

LLM-As-A-Judge evaluations and (4) community adoption metrics. Our re-analyses of the evidence lead us to conclude that the paper's own evidence invalidates its central claim: `min-p` sampling improves neither quality, nor diversity, nor the trade-off between quality and diversity.

Although Nguyen et al. (2024) presents a high-visibility example of scientific mistakes, the errors made in evaluating `min-p` are common in empirical machine learning research. Towards promoting rigorous science across the field, this case study offers several general lessons for researchers and reviewers alike. First, we develop a novel methodology for fairly comparing methods that require extensive hyperparameter tuning, which can also be used to detect potential cherry-picking. Second, we highlight best practices for statistical testing in human evaluations, particularly concerning the need to correct for multiple comparisons. Third, we demonstrate the critical role of data transparency and exploratory data analysis in uncovering flaws like omitted data and mischaracterized feedback. Ultimately, we argue that by establishing and adhering to robust methodologies, the field can better rein in questionable claims and ensure scientific progress advances.

## 2    RE-ANALYZING MIN-P'S HUMAN EVALUATIONS

We begin our case study by analyzing Nguyen et al. (2024)'s human evaluations, since human judgments are widely considered the gold standard for assessing language models (Van Der Lee et al., 2019; Roller et al., 2020; Howcroft et al., 2020; Clark et al., 2021; Liang et al., 2022; Khashabi et al., 2022; Chiang et al., 2024; Biderman et al., 2024; Schaeffer et al., 2025d). Prior work has exhaustively demonstrate human evaluations of language models frequently are laden with flaws Freitag et al. (2021); Belz et al. (2021); Thomson et al. (2024): We identified four issues.

### 2.1    HUMAN EVALUATORS' SCORES FOR ONE OF TWO BASELINE SAMPLERS WERE OMITTED

Section 6 of Nguyen et al. (2024) states human participants evaluated `min-p` against a single baseline sampler, `top-p`, and their Table 4 presents results only these two samplers. However, when examining the paper's data, we discovered that **scores for a second baseline sampler (`basic` sampling) were excluded from the methodology, the analysis and the results without mention or justification**. We publicly confirmed with the authors. These omitted scores comprised $1/3^{\text{rd}}$ of the total collected scores. After we raised the issue, the omitted data were added to the Camera Ready's Table 4, but the methodology, the results and the conclusions were not correspondingly updated. As we shall show, the omitted human evaluations data change the paper's conclusions.

### 2.2    VISUALIZATIONS AND STATISTICAL TESTS FAIL TO SUPPORT CLAIM THAT MIN-P OUTPERFORMS OTHER SAMPLERS

Based on the human evaluators' scores, Section 6 of Nguyen et al. (2024) concluded that `min-p` "consistently" outperformed `top-p` "across all settings":

| Metric | Alt. Hyp. | Temperature | | | | | |
|---|---|---|---|---|---|---|---|
| | | $\tau = 1.0$ | | $\tau = 2.0$ | | $\tau = 3.0$ | |
| | | $t$ | $p$ | $t$ | $p$ | $t$ | $p$ |
| Quality | Min-p $>$ Basic | 0.33 | .370 | 0.65 | .260 | $3.13^{***\dagger}$ | .001 |
| | Min-p $>$ Top-p | $2.05^{*}$ | .023 | 1.18 | .121 | $2.02^{*}$ | .025 |
| Diversity | Min-p $>$ Basic | 0.31 | .378 | $1.86^{*}$ | .034 | 0.85 | .201 |
| | Min-p $>$ Top-p | $2.64^{**}$ | .006 | 1.44 | .078 | 0.87 | .195 |

$^{*}\ p < 0.05$, $^{**}\ p < 0.01$, $^{***}\ p < 0.001$, $^{\dagger}$ Significant after Bonferroni correction for 12 comparisons.

Note: All tests were paired t-tests with df = 52, one-sided (alternative = "greater")

Table 1: **Hypothesis Testing of Human Evaluators' Scores Fails to Support Claim that `Min-p` Consistently Outperforms Other Samplers.** To test whether evidence supports the claim that `min-p` "consistently outperforms" other samplers, we conducted one-sided paired t-tests using the authors' published data. Without correcting for multiple comparisons, evidence supports `min-p`'s superiority in 5 of 12 comparisons at $\alpha = 0.05$ and 2 of 12 comparisons at $\alpha = 0.01$. After applying a Bonferroni correcting for multiple comparisons, evidence supports `min-p`'s superiority in 1 of 12 comparisons at $\alpha = 0.05$ and 0 of 12 comparisons at $\alpha = 0.01$. For details, see Sec. 2.3.

> "Overall, `min-p` sampling consistently scored higher than `top-p` sampling across all settings [...] A paired t-test confirmed that the differences in scores between min-p and top-p sampling were statistically significant ($p < 0.05$)."

However, **both visualizations and statistical hypothesis tests of the original human evaluation data suggest `min-p` is indistinguishable from the baselines in almost all settings.**

To briefly explain the human evaluation methodology, three samplers (`basic`, `top-p` and `min-p`) were compared in six conditions: three temperatures $(1.0, 2.0, 3.0)$ and two diversity settings ("high" and "low") corresponding to different $p$ hyperparameters. Humans were asked to score the generated outputs under two metrics: quality and diversity. Participants were excluded if they failed attention checks. For more information, please see the original manuscript.

We focused on the "high" diversity setting for three reasons: (1) The claimed advantage of `min-p` is that it provides both high quality and high diversity, whereas other samplers typically trade one against the other. (2) The authors publicly told us to focus on the high diversity setting, writing, "the low [diversity] settings were quite experimental." (3) We believe `top-p`'s $p$ hyperparameter in the low diversity setting was poorly chosen; indeed, after we raised said concerns, the authors ran a new human evaluation (more in Sec. 2.4) which changed the low diversity `top-p` $p$ from $0.1 \rightarrow 0.9$.

**Using the original paper's data, Fig. 1 reveals that the three samplers provide similar quality and similar diversity, with 95% confidence intervals frequently overlapping**. To more rigorously assess the claim that `min-p` consistently outperforms other samplers, we conducted 12 one-sided paired t-tests for each metric (quality or diversity), temperature $(1.0, 2.0, 3.0)$ and baseline sampler (`min-p` versus `basic`, `min-p` versus `top-p`). In each test, the null hypothesis is `min-p`'s score is less than or equal to the other sampler's score, and the alternative hypothesis is `min-p`'s score is greater than the other sampler's score. Statistical test results are displayed in Table 1. Without correcting for multiple comparisons, we found evidence to reject the null hypotheses in 5 of 12 tests at $\alpha = 0.05$ and 2 of 12 tests at $\alpha = 0.01$. After applying a Bonferroni correction for multiple comparisons, we found evidence to reject the null hypothesis in 1 of 12 tests at $\alpha = 0.05$ and 0 of 12 tests at $\alpha = 0.01$. Furthermore, given that the original paper claims that `min-p` "consistently" scores higher, an Intersection-Union Test (IUT) may be the appropriate statistical test, where the alternative hypothesis is that `min-p` is better in all 12 comparisons and the null hypothesis is the set complement. Since the largest p-value of the 12 comparisons is $0.378$, under the IUT, we again find insufficient evidence to reject the null hypothesis at both $\alpha = 0.05$ and $\alpha = 0.01$. **Based on the original paper's data, there is insufficient evidence to support the claim that `min-p` consistently outperforms baseline samplers across all settings.**

The original paper's statistical analysis reached an incorrect conclusion due to a combination of two factors: First, despite claiming that `min-p` "consistently scored higher" "across all settings"

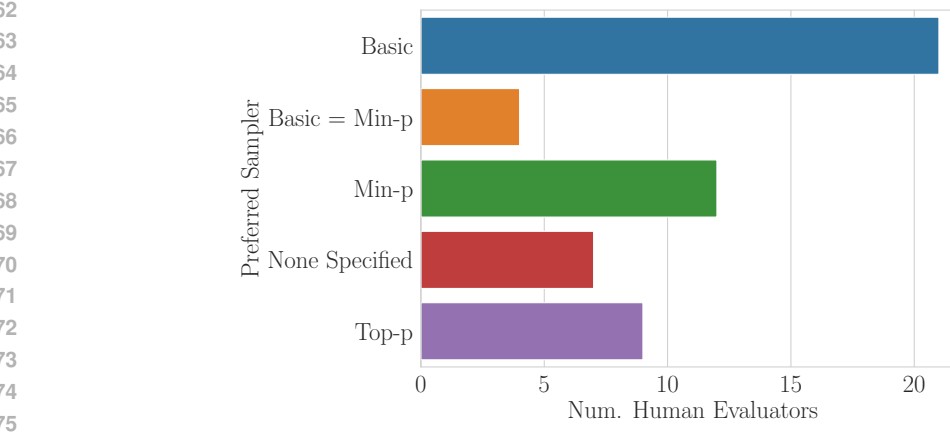

Figure 2: **Manual Annotation of Human Evaluators' Qualitative Responses Fail to Support Claim that `Min-P` Was the Preferred Sampler.** We manually annotated responses from human annotators regarding their preferred sampler(s) at the end of the original paper's study. The responses suggest `min-p` was not the most preferred sampler. Example responses are provided in Appendix. B.

(metric, temperature, and diversity), the paper pooled data across all settings and performed a single t-test, which tests whether `min-p` scored higher on average. Second, pooling over all settings was misleading because `top-p`'s hyperparameter $p$ was poorly chosen in the "low" diversity condition in a way that pulled `top-p` down significantly; the authors said publicly to ignore this particular low diversity condition and subsequently changed $p$ in their new human experiment (Sec. 2.4).

## 2.3 Human Evaluators' Qualitative Responses Fail to Support Claim That Min-P Is Preferred Over Other Samplers

At the end of the human evaluation study, the original paper asked human participants to qualitatively describe which sampler(s) they preferred. The paper claimed that human evaluators' qualitative responses support `min-p` over `top-p`:

> "Participants frequently noted that outputs generated with min-p sampling were more coherent and creative, especially at higher temperatures."

However, we believe that **the paper's qualitative human responses contradict this**. We manually annotated the qualitative responses, then visualized our annotations of the humans' expressed preferences (Fig. 2) and publicly posted our annotations in the same format. We found two results: (1) more human evaluators explicitly preferred `basic` sampling than preferred `min-p` sampling, which was not immediately obvious because the `basic` sampling data were previously excluded (Sec. 2.1), and (2) `min-p` was only slightly preferred over `top-p`. We provide quotations from human evaluators favoring `basic` sampling in Appendix B.

## 2.4 New Human Evaluation Study Shows Min-P Does Not Outperform Baselines in Quality, in Diversity, or in a Tradeoff Between Quality and Diversity

In response to our feedback, the authors conducted and added a new human evaluation study to Appendix C.2. Their new study made multiple methodological changes:

- Different sampler implementation: switched from applying temperature *after* truncation to applying temperature *before* truncation.
- Different distribution of human participants from Prolific.
- Different hyperparameters for `top-p`: switched from 0.1 and 0.9 to 0.9 and 0.95.
- Different hyperparameters for `min-p`: switched from 0.2 and 0.05 to 0.1 and 0.05.
- Different allotted reading time: increased from 30 minutes to 45 minutes.

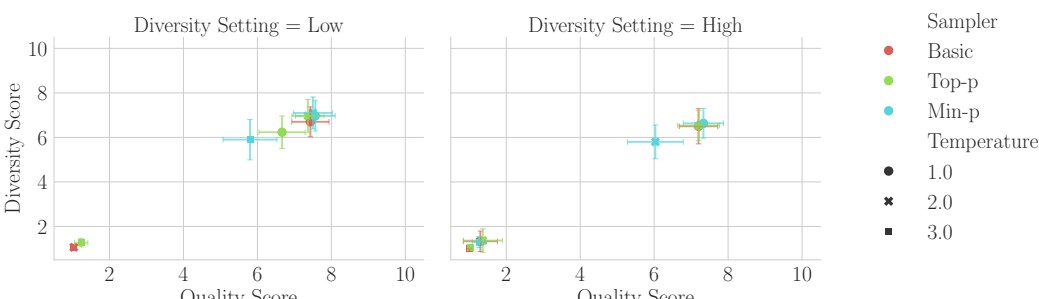

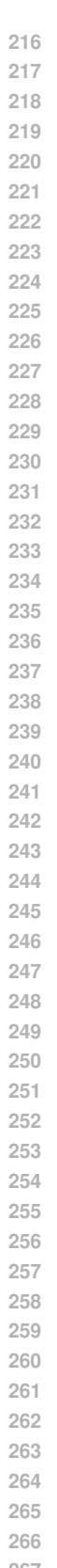

Figure 3: **New Human Evaluation Study Suggests `Min-p` Does Not Outperform Baselines in Quality, in Diversity or in a Pareto-Optimal Tradeoff Between Quality and Diversity.** Visualization of scores from Nguyen et al. (2024)'s second human experiment. For researchers or practitioners seeking maximal quality and maximal diversity, these human evaluation results suggest that `min-p` offers no apparent advantage over `basic` sampling or `top-p` sampling.

- Different sampled text: 3 short paragraphs were replaced with a single complete story.
- Different rubric for human participants to evaluate sampled outputs.

Regarding the new human evaluation data and results, we share two discoveries here: First, more generally, the data show again that `min-p` does not outperform baseline sampling methods in quality, in diversity or in a favorable tradeoff between quality and diversity (Fig. 3). Second, we believe one value is incorrectly reported: in Nguyen et al. (2024)'s Table 15, the average score of `min-p` at $p = 0.05$ and temperature $T = 2$ is reported as 7.80, but based on the authors' publicly posted data, we believe the correct numerical value should be 5.80. **For anyone seeking higher quality or diversity, `min-p` offers no apparent advantage over previously existing samplers**.

## 3 EXTENDING MIN-P'S NLP BENCHMARK EVALUATIONS

We next turned to the original paper's NLP benchmark evaluations of several models on GSM8K with Chain-of-Thought (Cobbe et al., 2021) and GPQA (5-shot) (Rein et al., 2023), which concluded that:

"`Min-p` sampling achieves superior performance across benchmarks and temperatures."

### 3.1 THOROUGH HYPERPARAMETER SWEEP ON GSM8K CONTRADICTS CLAIM OF MIN-P'S SUPERIORITY

Verifying whether `min-p` achieves superior performance was difficult because it was not clear whether sampling methods were compared equally. To test this, we used the authors' code to conduct an extensive sweep on GSM8K CoTover the following models, samplers and hyperparameters:

- **9 Models:** Qwen 2.5 (Qwen et al., 2025) 0.5B, 1.5B, 3B and 7B; Mistral 7Bv0.1 (Jiang et al., 2023); Llama (Grattafiori et al., 2024) 3.1 8B and 3.2 3B; Gemma 2 (Team et al., 2024) 2B and 9B.
- **2 Model Stages:** Pre-trained ("Base") and Post-Trained ("Instruct").
- **4 Samplers:** `basic`, `top-p`, `top-k`, `min-p`.
- **31 Temperatures:** 0.0 ("greedy") to 3.0 in increments of 0.1.
- **6 Hyperparameters Per Sampler:** We chose 6 hyperparameters per sampler, except for `basic` which has no hyperparameter beyond temperature. The values were taken from the original paper; some were lightly edited to make them more evenly distributed:
  - `basic`: No hyperparameters other than temperature.
  - `top-k`: $k \in \{10, 30, 50, 100, 150, 200\}$.
  - `top-p`: $p \in \{0.99, 0.98, 0.95, 0.9, 0.8, 0.7\}$.
  - `min-p`: $p \in \{0.01, 0.02, 0.05, 0.1, 0.2, 0.3\}$.

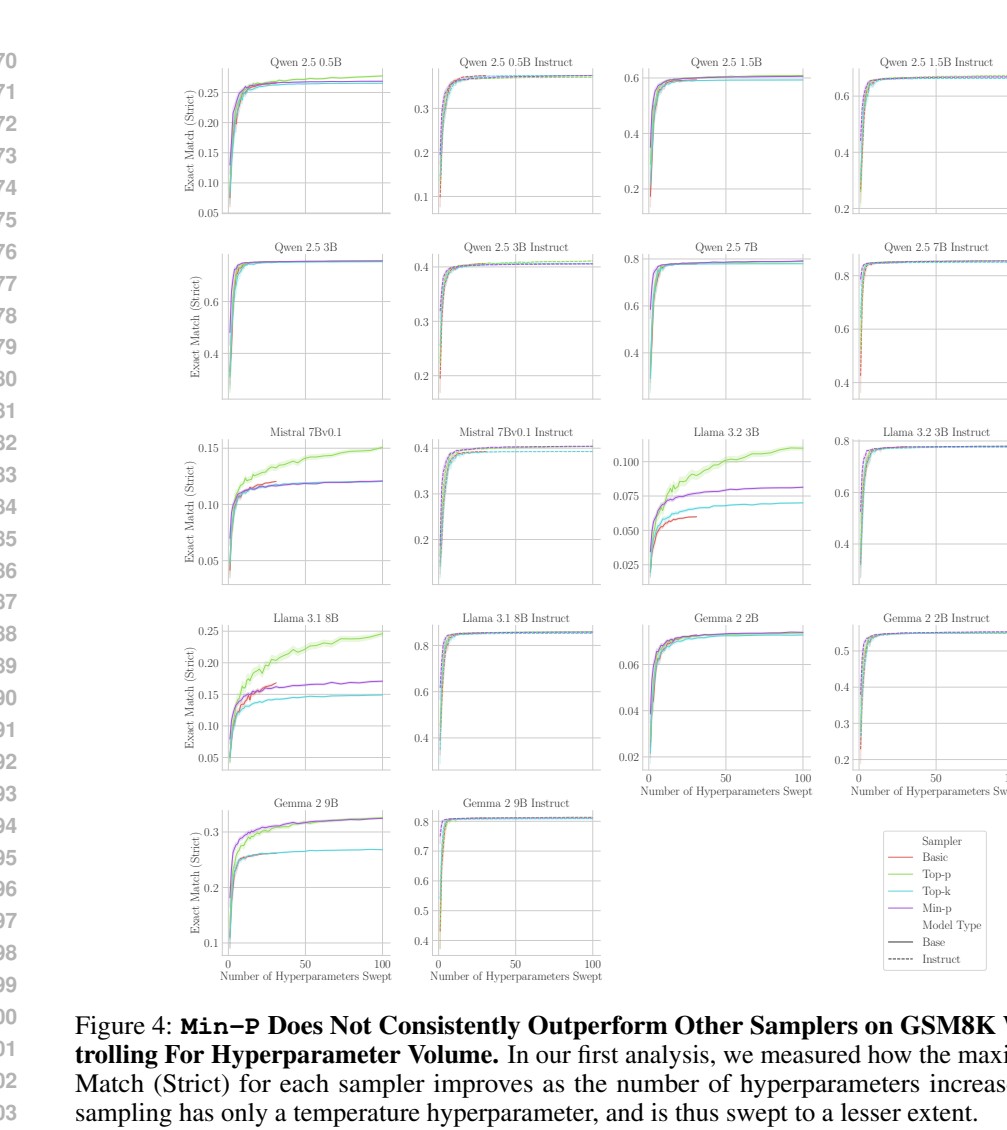

Figure 4: **Min-P Does Not Consistently Outperform Other Samplers on GSM8K When Controlling For Hyperparameter Volume.** In our first analysis, we measured how the maximum Exact Match (Strict) for each sampler improves as the number of hyperparameters increases. Basic sampling has only a temperature hyperparameter, and is thus swept to a lesser extent.

- **3 Random Seeds for Sampling:** $\{0, 1, 2\}$

Due to our compute budget, we only evaluated GSM8K CoT (albeit under two prompt formats, for reasons explained below). This sweep and the sweep below required $\sim 6000$ Nvidia A100-hours.

To evaluate how performant each sampler is, we first averaged over the three sampling seeds and then conducted two complementary analyses:

1. For each sampler, we subsampled an equal number of hyperparameters ranging from $N = 1$ to $N = 100$ and computed the maximum Exact Match (Strict) score achieved by the sampled subset of size $N$. We repeated this process 150 times, averaging over the subsampled subsets' scores. This "Best-of-N" analysis (Nakano et al., 2021; Stiennon et al., 2020; Hughes et al., 2024; Schaeffer et al., 2025c) tells us the best possible performance each sampler will likely obtain as its hyperparameter space increases.

2. For $N = 1$ to $N = 100$, we subsampled $N$ hyperparameters per sampler and computed the difference of the maximum Exact Match (Strict) score achieved by min-p minus the maximum score achieved by any other sampler. We repeated this process 150 times, averaging over the subsampled subsets. This tells us by how much min-p outperforms all other samplers, controlling for the size of hyperparameter space of each sampler.

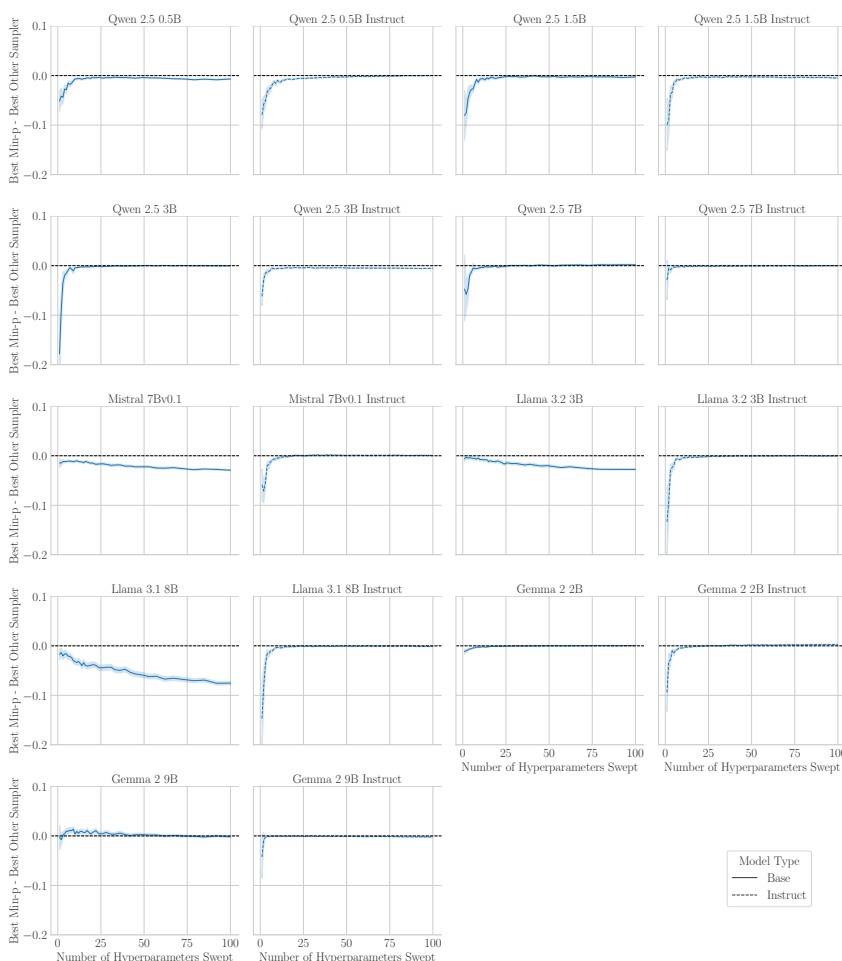

Figure 5: **Min-P Does Not Consistently Outperform Other Samplers on GSM8K When Controlling For Hyperparameter Volume.** In our second analysis, we measured how the difference between min-p's highest score and the best non-min-p sampler's highest score changes as the number of swept hyperparameters increases. Min-p matches or underperforms other samplers.

**Both analyses reached consistent results: min-p does not outperform other samplers when equalizing the volume of hyperparameter space.** Fig. 4 and Fig. 5 respectively demonstrate that min-p is largely indistinguishable from other samplers. After we showed these results to the authors, they informed us that their code default used the incorrect benchmark prompt formatting. We reran the experiments using standard formatting of GSM8K CoT prompts. The results were nearly identical (Appendix C), with one small difference: min-p does produce higher scores for 2 of 12 language models. Min-p does not outperform other samplers when controlling for hyperparameter volume.

## 4    INVESTIGATING MIN-P'S LLM-AS-A-JUDGE EVALUATIONS

We next turned to the original paper's LLM-as-a-Judge evaluations (Zheng et al., 2023), specifically AlpacaEval creative writing evaluations (Dubois et al., 2023).

### 4.1    UNDER-SPECIFIED AND INDIRECT METHODOLOGY HINDERS REPRODUCTION AND INTERPRETATION

The methodology in the manuscript was under-specified in several ways: There is no mention which model(s) were sampled from, which model(s) served as the judge(s), or how hyperparameters were chosen or swept. Additionally, there is no description of uncertainty for reported win rates, meaning

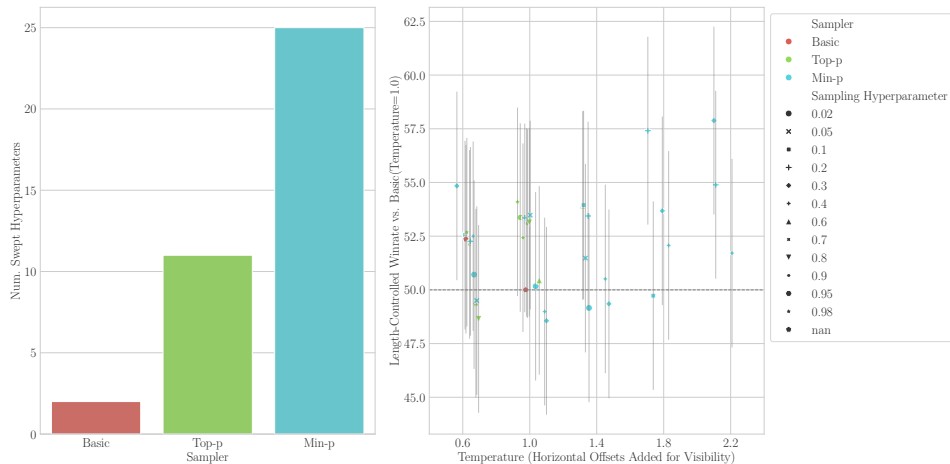

Figure 6: **Nguyen et al. (2024)'s LLM-As-A-Judge Evaluations Suggest `Min-p` Typically Matches Other Samplers Despite** $2\times$ **to** $10\times$ **More Hyperparameter Tuning.** Left: Nguyen et al. (2024) swept `min-p` with more than twice as many hyperparameters as `top-p` and more than ten times as many hyperparameters as `basic`. Right: Pairwise comparisons show `min-p` typically performs on-par with other samplers. Data were obtained from a public GitHub repository.

win rates may be indistinguishable from chance (50.00%). Furthermore, the experiment seems designed in a manner that introduces a confounder. As context, AlpacaEval reports win rates between pairwise comparisons. Instead of directly comparing `min-p` against other samplers, the authors compared each sampler against a common fixed sampler: `basic`($\tau = 1.0$). This comparison strategy is indirect; comparing directly against `min-p` would offer a clearer assessment of its superiority while using the same number of comparisons. The experimental design is additionally concerning because LLM-judge preferences are not transitive, as shown by recent research (Xu et al., 2025). Therefore, comparing all methods to `basic`($\tau = 1.0$) provides no reliable inference about `min-p`'s performance relative to `top-p` or `basic` at other temperatures. **These under-specified aspects of the methodology and the indirect experimental design make drawing conclusions difficult.**

### 4.2 Min-p Received More Hyperparameter Tuning Yet Frequently Fails to Win

Closely scrutinizing (ongoing work to publish) the data revealed two more insights: (1) `min-p` received $\sim 2\times$ more hyperparameter tuning than `top-p` sampling and $\sim 10\times$ more tuning than `basic` sampling (Fig. 6, left), potentially tilting the scales in its favor. (2) the win-rates show that `min-p` frequently fails to outperform `top-p` and `basic` sampling, especially when accounting for confidence intervals; we visualized the new data with 95% confidence intervals (with horizontal offsets added for visibility) (Fig. 6, right).

### 4.3 Table 3(b) Reported The Higher of Two Scores For Min-p But the Lower of Two Scores For Top-p

As evidence for the LLM-As-A-Judge evaluation scores in the original paper's Table 3(b), the first author publicly shared a Telegram link that showed the higher of two scores was reported for `min-p` (the reported win rate of 52.01 corresponds to $p = 0.05$, but $p = 0.01$ yields a lower win rate of 50.14) but the lower of two score was reported for `top-p` (the reported win rate of 50.07 corresponds to $p = 0.9$, but $p = 0.98$ yields a higher win rate of 50.43).

### 5 Min-p's Claimed GitHub Repositories & Stars Were Unsubstantiated and Retracted

Nguyen et al. (2024) included specific claims about `min-p`'s widespread adoption:

"**Community Adoption:** Min-p sampling has been rapidly adopted by the open-source community, with over 54,000 GitHub repositories using it, amassing a cumulative 1.1 million stars across these projects."

We attempted to verify these numbers through analysis of major GitHub language modeling repositories. Per our calculations, the combined GitHub stars of leading LM repositories (`transformers`, `ollama`, `llama.cpp`, `vLLM`, `Unsloth`, `mamba`, `SGLang`, `llama-cpp-python`) sum to 453k stars as of March 2025, less than half the 1.1M stars claimed by `min-p` alone. We could not substantiate either 49k GitHub repositories or 1.1M GitHub stars. When we inquired how these numbers were calculated, the authors publicly stated that GitHub was searched for "min-p", which yields many false positives. **The authors retracted both the 54k GitHub repository claim and the 1.1M GitHub stars claim from the ICLR 2025 Camera Ready manuscript.** We highlight this point because 3 of 4 ICLR 2025 reviewers and the Area Chair identified these retracted community adoption numbers as the main justification for their strong endorsement. The ICLR 2025 Camera Ready manuscript has a different statement of community adoption, which we believe remains misleading.

## 6 DISCUSSION AND LIMITATIONS

**Scientific Conclusions:** Our case study led us to conclude that the four lines of evidence presented by Nguyen et al. (2024) – (1) human evaluations, (2) NLP benchmark evaluations, (3) LLM-as-a-Judge evaluations, (4) community adoption – do not support `min-p`'s claimed superiority. While `min-p` is useful as another method to try, the original paper's data and our extensions of the original paper's data suggest that samplers perform approximately equally if given equal hyperparameter tuning.

**Key Limitation:** Our manuscript re-analyzes the evidence presented by Nguyen et al. (2024) and additional evidence created using the original paper's code. *Conclusions here are based on that evidence.* We emphasize that new evidence might lead to different conclusions.

**General Lessons for Reviewers and Researchers:** This case study of `min-p` reveals several general lessons for more rigorous science in empirical machine learning research:

1. **Compare methods fairly by controlling for hyperparameter volume.** As demonstrated in the NLP benchmark analysis, a method's advantage may disappear when the volume of hyperparameter space being searched is equalized across all contenders. Our "Best-of-N" analysis is an effective way to control for this and detect potential cherry-picking.

2. **Apply statistical tests rigorously and transparently.** The re-analysis of human evaluation data reveals how incorrect statistical practices can lead to false conclusions. This includes inappropriately pooling data across different experimental conditions, and, failing to correct for multiple comparisons when testing multiple hypotheses. Visualizing data with appropriate uncertainty estimates is another crucial step to prevent misinterpretation.

3. **Demand and practice full data transparency.** A key finding was that the original study omitted one-third of its human evaluation data without justification, and the inclusion of this data changed the paper's conclusions. To allow for independent verification, researchers should release all collected data, annotations, and analysis code.

4. **Scrutinize qualitative summaries and unrealistic claims.** The original paper's summary of qualitative feedback did not align with a direct reading of the evaluators' responses. Furthermore, bold claims regarding community adoption were unsubstantiated, yet they heavily influenced the original reviewers' assessments Such claims must be carefully verified.

5. **Ensure methodological clarity for full reproducibility.** The LLM-as-a-Judge evaluations lacked crucial details hindering interpretation and reproduction. Empirical papers must provide enough detail for other researchers to faithfully replicate the work.

6. **Watch for inconsistent or selective reporting.** In the LLM-as-a-Judge results, the higher of two scores was reported for min-p, while the lower of two was reported for a baseline method. This selective reporting creates a misleading picture of the method's performance. All results must be reported using a consistent methodology to avoid bias.

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

## A  LANGUAGE MODEL USAGE

Language models were used by the authors to aid or polish the writing of the paper. Authors take full responsibility for the content.

## B    EXAMPLES OF HUMAN QUALITATIVE RESPONSES FAVORING BASIC SAMPLING OVER MIN-P SAMPLING

In Section 2.3, we described how qualitative responses from many human participants in the original paper's study favored basic sampling. Direct quotes from human evaluators favoring basic sampling are provided below. In the study, basic sampling was called "Model A"; for clarity, we substituted the pseudonyms for the actual sampling methods):

- "[basic sampling] on Temp 3.0 - High Diversity setting. The stories where [sic] more interenting [sic], felt more different compared to the others, which felt like the same ideia [sic] just in a different format."

- "I felt like [basic sampling] was most diverse and most interesting with it's [sic] descriptions of the characters and the setting. It appealed to me most and seemed to have less 'broken' sentences that didn't make sense. Descriptions were painterly [sic] and elaborate."

- "[basic sampling] was more engaging, it aroused my curiosity."

- "[basic sampling] provided more depth and easy to read for me and there was more diversity."

- "[basic sampling], they presented creative storytelling"

- "[basic sampling]. From the very beginning the verbiage and descriptions were very creative and vivid. And each story was unique"

- "I believe that [basic sampling] has provided stories with more differentiation overall than the other two models. From the point of view of creativity, all three models are more or less equivalent as they almost always talk about stories set in extraterrestrial worlds both from a physical and mental (dreams) point of view"

- "[Basic sampling]: Sample 2: Temperature Setting F (Temp 3.0 - High Diversity). The story was captivating, it took inside the mystical land and walked you right besides all the characters, you can even draw the characters from just th descriptions provided by the prompt. you Could even smell them, smell the setting and be at one with the setting."

- "I personally preferred [basic sampling] on the setting of creative, descriptive storytelling. I enjoyed how the writing was creative, showing imagination and a strong use of language. The stories were quite evocative, with intriguing settings and characters that helped to draw the reader in. I also appreciated the diversity of themes that were explored, from night weavers to dream manipulation and mysterious libraries, which kept the stories engaging and interesting."

- "Temperature setting C on [basic sampling] was the best. The story was fascinating and very engaging. I wanted to read more."

- "I prefered the first [basic sampling]. Tho [basic sampling] and C seem to be very head to head. But something about [basic sampling] seemed different in quality about it to me."

More quotes are in the original paper's data. We urge readers to draw their own conclusions.

## C    GSM8K Chain-of-Thought Scores with "Standard" Formatting

At the request of Nguyen et al. (2024), we reran our GSM8K Chain-of-Thought sweeps using "standard" formatting instead of "Llama" formatting. **Both analyses reached consistent results: `min-p` does not consistently outperform other samplers when controlling the volume of hyperparameter space.**

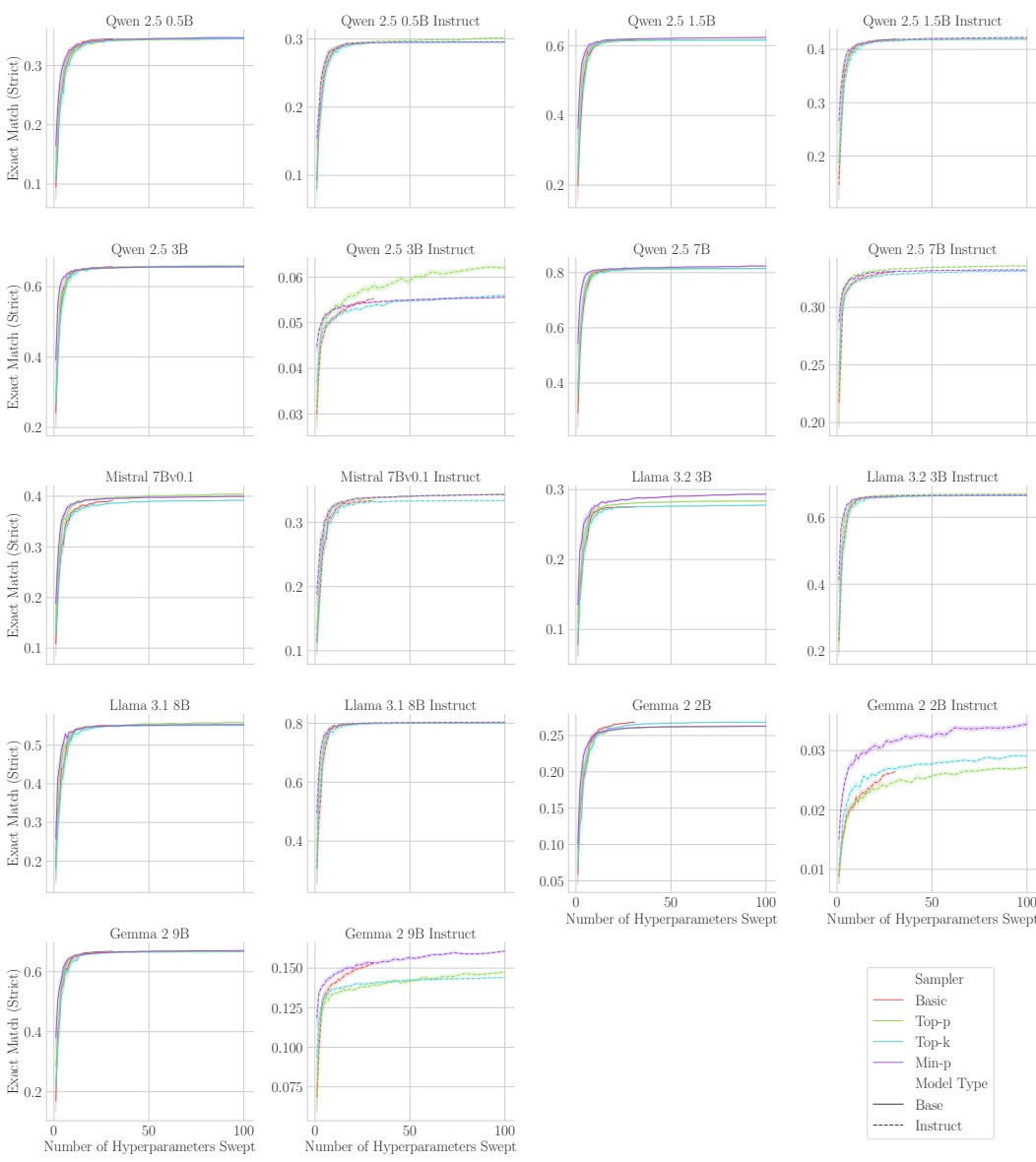

Figure 7: **`Min-P` Does Not Consistently Outperform Other Samplers on GSM8K When Controlling For Hyperparameter Volume.** We reran our GSM8K sweep using "standard" formatting rather than "Llama" formatting and observed qualitatively similar data.

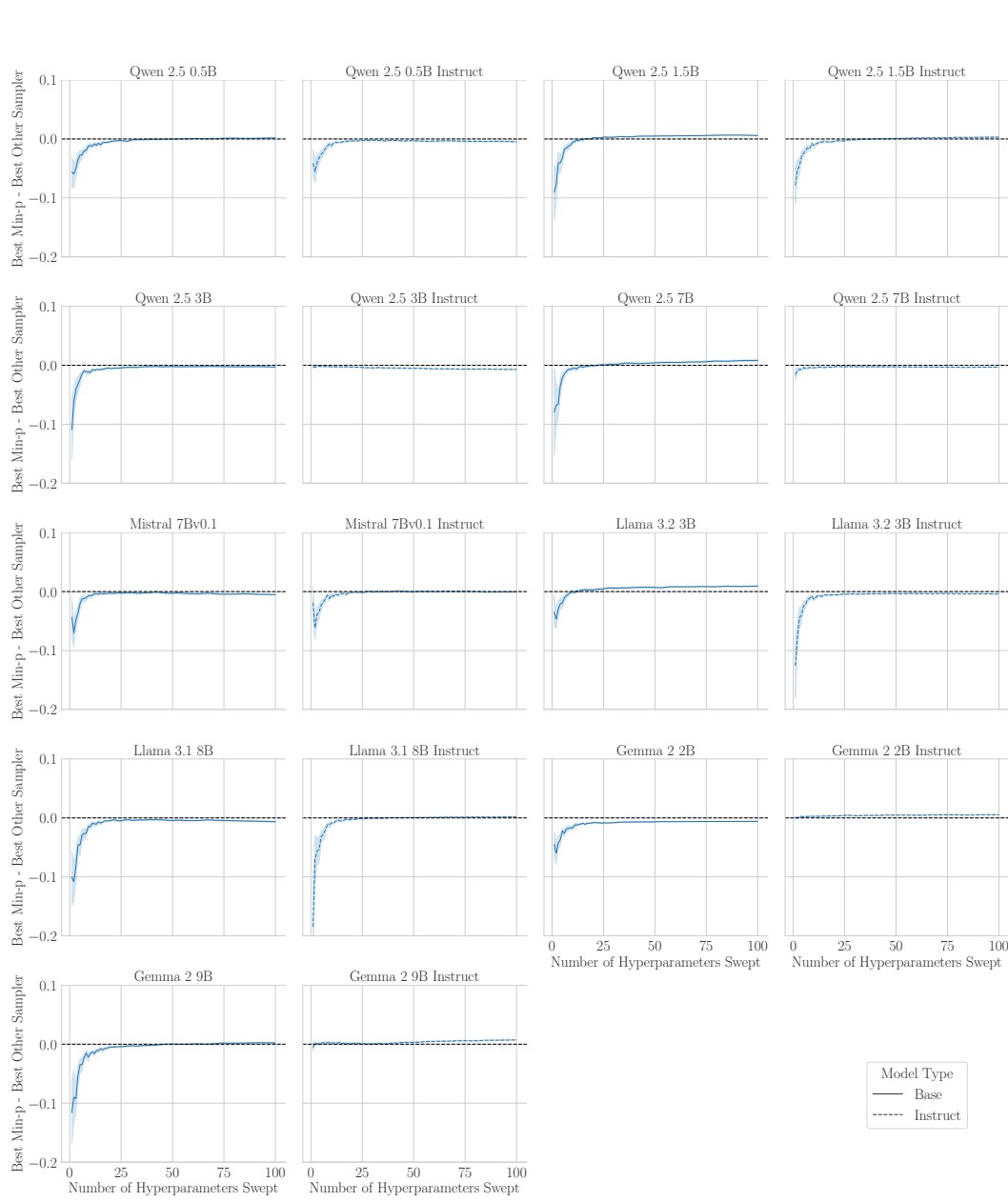

Figure 8: **Min-P Does Not Consistently Outperform Other Samplers on GSM8K When Controlling For Hyperparameter Volume.** We reran our GSM8K sweep using "standard" formatting rather than "Llama" formatting and observed qualitatively similar data.

# D    GSM8K Scores By Model, Sampler and Hyperparameters

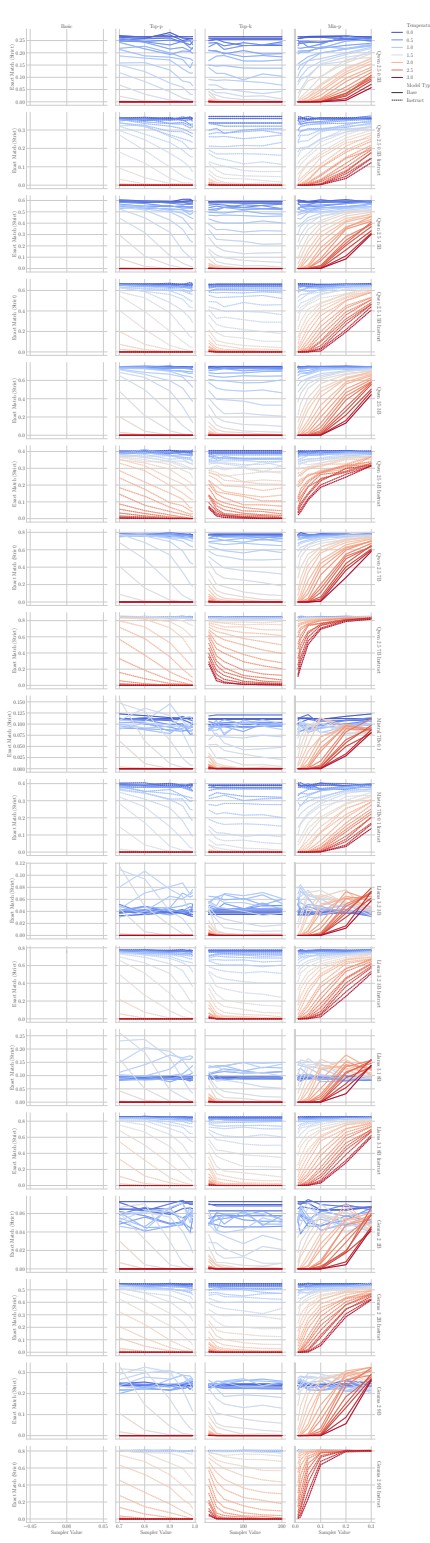

Figure 9: **GSM8K Scores By Model, Sampler and Sampler Hyperparameters.** Many models achieve their highest scores at low temperatures across samplers.

