# OpenReview forum: "A $\texttt{Min-p}$ Blueprint for More Rigorous Science in Empirical Machine Learning Research"
_ICLR.cc/2026/Conference — Submitted to ICLR 2026_

### Official Review · Reviewer_2vSc · 2025-10-19

**Soundness:** 2
**Presentation:** 2
**Contribution:** 1
**Rating:** 2
**Confidence:** 4

**Summary:**

This work challenged the experimental results of the min-p method (ICLR 2025 oral), claiming that most of its conclusions are invalid upon closer examination. The paper ends with a list of general lessons for conducting more rigorous empirical ML studies.

**Strengths:**

- A comprehensive re-examination of a seemingly popular method for token sampling in LLMs

- While most of the key lessons pointed by the authors are not new per se in established guidelines on reproducibility, it is nonetheless a good case study to draw more attention to the current questionable practices in ML research

**Weaknesses:**

- This work seems to be an odd fit for ICLR, as there is no new result or particularly novel insights. Perhaps the authors should consider submitting this work to the ML reproducibility challenge.

- Some of the claims (eg., S2.1, S2.4, S4.3), including private communications to the original authors of min-p, are difficult to verify for the reviewers.

- Since the work is mostly focused on criticizing one particular paper, it feels only just if the original authors were given a chance to respond before publication. However, I do not see how the existing ICLR protocol could accommodate this exchange.

- I agree with the authors that there is a crisis of rigor in empirical ML research. This is a widespread problem in the entire community (sometimes caused by realistic constraints such as resource). However, I do not think it is a good idea to "go after" one particular paper or group. This work would be much more useful and impactful if it is organized by the current ill practices in empirical ML (like the lessons that the authors listed on Line 28-30), with each lesson illustrated on a **different** high profile paper. We need to distinguish criticizing a paper/group (more subjective and not interesting to the community) from criticizing a practice (more objective and interesting to everyone).

**Questions:**

Let me be clear: I applaud the authors' efforts and I agree with the listed lessons (though I can't say any of them is really new or unexpected). I just do not think ICLR is the right place for such efforts, and I prefer not to single out any paper or group when the underlying problem is widespread.

The disclaimer on Line 456 (new evidence might lead to different conclusions) makes me feel even more uneasy, if the original authors of min-p were not given a chance to respond. But I don't know how this could be accommodated.

**Details Of Ethics Concerns:**

I am not sure if this type of reproducibility study is a right fit for ICLR. Given that the publication of this work could be inflammatory to the original authors of min-p, it only feels just if the original authors were given a chance to respond. But how?

Also, some of the claims in this work are difficult for a reviewer to verify (e.g., private communications).

---

> ### Author Response · Authors · 2025-11-20
> **Response to Reviewer 2vSc**
>
> We thank Reviewer 2vSc for recognizing our work as a "comprehensive re-examination" and agreeing that there is a "crisis of rigor" in the field. We take the reviewer's concerns regarding ethics and fairness very seriously and address them below.
>
> # Response to Fairness, Ethics, and the "Right of Reply" (Re: Weakness 3 & Ethics Flag)
>
> The reviewer expresses concern that criticizing a single paper feels "unfair" if the authors cannot respond. We wish to clarify three crucial points:
>
> - The Authors Did Respond: As noted in our manuscript (Sections 2.1, 2.4, 5), we engaged in extensive communication with the original authors prior to writing this paper. In fact, the "New Human Evaluation" (Section 2.4) and the retractions of the GitHub star claims (Section 5) were performed by the original authors in response to our findings. This manuscript is not a "blind attack"; it is the culmination of multiple rounds of scientific dialogue.
>
> - Post-Publication Review is Standard Science: In mature scientific fields, publishing a paper that invalidates a prior high-profile claim is standard practice (e.g., the "arsenic life" controversy in biology or the replication crisis in psychology). "Going after" a specific claim is not a personal attack; it is the mechanism by which science corrects itself.
>
> - Why Single Out One Paper? We focused on min-p because a broad survey cannot go deep enough to expose these specific types of errors. To find the statistical flaws in the human eval and the hyperparameter volume issues, we had to spend ~6,000 GPU-hours and manually annotate qualitative data. A survey of 10 papers would have been superficial. Furthermore, min-p was the 18th highest ranked paper at ICML 2025, making it a representative case study of the highest visibility.
>
> # Response to Weakness #1: Venue Fit and the Reproducibility Challenge
>
> The reviewer suggests the "ML Reproducibility Challenge." We respectfully disagree.
>
> - Reproducibility vs. Validity: The Reproducibility Challenge largely asks: "Does the code run and produce the reported number?" Our paper asks: "Is the scientific conclusion valid?" We successfully ran their code (reproducibility), but we proved the method provides no benefit when tuning is controlled (validity).
>
> - Precedent: ICLR and NeurIPS regularly publish papers that scrutinize single topics or methods to correct the record. Perhaps the most famous example includes Engstrom et al. ICLR 2020 Oral “Implementation Matters in Deep Policy Gradients: A Case Study on PPO and TRPO”, which is explicitly a “case study”. Two more examples are Haibe-Kains et al. (2020) in Nature on shortcomings of a previous paper and Carlini et al. (2020)’s “Is Private Learning Possible with Instance Encoding?” which specifically challenges a previous paper (InstaHide).
>
> When an ICLR Oral omits data, conducts improper statistical tests, draws invalid conclusions, and fabricates community adoption figures, then ICLR has an obligation to host a correction.
>
> # Response to Weakness #2: Verifiability of Claims
>
> The reviewer notes that private communications are hard to verify. However, the artifacts of these communications are public and verifiable. The reviewer can verify these changes by manually stepping through the history of revisions that ICLR makes publicly available through OpenReview: https://openreview.net/revisions?id=FBkpCyujtS. The following are easily verifiable:
>
> Omitted Data: The original authors updated their Camera Ready Appendix to include the previously omitted baseline scores (after we flagged it).
>
> Retractions: The original authors removed the claimed 1.1M GitHub stars and 50,000 GitHub repositories  claim from their Camera Ready version (after we flagged it).
>
> We would be happy to provide direct links to specific versions with an annotated chronology, if our doing so would be helpful?
>
> # Response to Weakness #3: Concern with Disclaimer
>
> The reviewer felt uneasy about our statement: "new evidence might lead to different conclusions."
>
> This is a standard statement of epistemic humility. In empirical science, one can never prove a negative with 100% certainty. However, after 6,000 GPU hours and rigorous statistical re-analysis, the burden of proof has shifted back to the proponents of min-p.
>
> # Conclusion
>
> We believe that rigorous post-publication review is essential for the health of the ML community. We hope this response clarifies that our process was ethical, communicative with the original authors, and necessary for correcting the scientific record.

---

> > ### Comment · Reviewer_2vSc · 2025-11-24
> > **Re: Response to Weakness 1**
> >
> > The authors' description of the reproducibility challenge is not accurate. Let me copy the call for paper from MLRC'25 (https://reproml.org/call_for_papers/):
> > - Methods and tools to foster reproducibility research in Machine Learning
> > - Generalisability of published claims: novel insights and results beyond what was presented in the original paper, from any paper (or set of papers) published in top ML conferences and journals.
> > - Meta-reproducibility studies on a set of related papers.
> > - Meta analysis on the state of reproducibility in various subfields in Machine Learning.
> >
> > Note in particular point 2 and point 4.
> >
> > It is interesting that the authors brought up Engstrom et al. (2020) and Carlini et al. (2020). Now anyone can read and judge by themselves on the difference between these studies and this submission. To me: this paper focused on rebuttal each claim made in min-p, without any of its own novel contributions. This is not the case for Engstrom et al. (2020) and Carlini et al. (2020): they each contained novel lessons beneficial to the community, and they do not read as mere rebuttals against a previous work.
> >
> > [I am ignoring Haibe-Kains et al. (2020) since I am fairly certain journals operate differently: the editor must have consulted the original authors (not necessarily agreeing with them).]
> >
> > I agree that the original authors, and to some extent ICLR, have an obligation to correct any wrong claims. However, this needs to be done prudently, which I do not think can be reasonably achieved with 3 blind reviewers in 2 weeks (while these reviewers also need to review 4,5,6,7... other papers). It is unrealistic to expect a blind reviewer to independently cross-examine every prior exchange and evidence. Reviewers are not even encouraged to check arxiv (to respect anonymity).
> >
> > I note that all reviewers seem to agree ICLR may not be the best venue for this submission. Maybe we are old-fashioned, or maybe the authors should rethink.

---

> > > ### Author Response · Authors · 2025-11-25
> > > **Comment on Novelty, Venue, and Ethics**
> > >
> > > We thank Reviewer 2vSc for the continued engagement. We appreciate the reviewer's concern for fairness and their rigorous standard for ICLR. However, we respectfully disagree with the assessment regarding novelty and the feasibility of peer review in this context.
> > >
> > > ## This work provides a novel methodological contribution for fair comparisons (Best-of-N)
> > >
> > > The reviewer states that Engstrom et al. (2020) contained "novel lessons" while our work is a "mere rebuttal." We strongly contest this.
> > > Engstrom et al. demonstrated that "code-level optimizations" confound RL evaluations. Similarly, our paper demonstrates that "hyperparameter tuning volume" confounds LLM sampler evaluations.
> > > We do not simply re-run min-p; we introduce a formal "Best-of-N" methodology (Section 3, Fig 4 & 5) to control for the size of the search space. This is a generalizable methodological contribution that solves a widespread problem in the community: new methods often look better simply because they are tuned more aggressively than baselines. This insight and the accompanying methodology are exactly the type of "novel lesson" the reviewer rightfully praises in Engstrom et al.
> > >
> > > ## Why MLRC is insufficient for an ICLR correction.
> > >
> > > While the MLRC is a valuable initiative, its primary focus is verifying if code runs and claims are reproducible. Our work goes beyond reproducibility; we are proving that the conclusions derived from the data are invalid.
> > >
> > > Furthermore, the paper we analyze (min-p) was an ICLR Oral. If ICLR publishes a method that is statistically indistinguishable from baselines—yet claims superiority based on flawed human eval and retracted adoption metrics—it is the responsibility of ICLR to publish the correction. Relegating the correction to a different, lower-visibility venue allows the scientific error to persist in the top-tier record.
> > >
> > > ## The “Damage” Acts Both Ways
> > >
> > > The reviewer notes the potential "damage" to the original authors. We ask the reviewer to consider the damage to the scientific community.
> > > The original paper claimed min-p was used by 1.1 million stars-worth of repositories—a claim that was false and retracted. Thousands of researchers and practitioners spend compute cycles and time implementing methods promoted as ICLR Orals. Publishing a rigorous invalidation prevents the community from wasting resources on a method that offers no advantage when properly tuned.
> > >
> > > To share a personal story, our project began by spending three months trying to improve upon min-p before discovering that any method could be made to look superior depending on how results were selectively presented. We are trying to correct the scientific record so that others don't have to pay a similar cost.
> > >
> > > ## Verification is Feasible
> > >
> > > The reviewer mentions that verifying these claims is difficult for them to do. We respectfully reiterate that the OpenReview revision history is public: https://openreview.net/revisions?id=FBkpCyujtS
> > >
> > > A quick glance at the "Revisions" tab confirms the addition of the omitted data and the removal of the GitHub star claims in the Camera Ready version.
> > >
> > > ## Response to Ethical Concerns and "Right of Reply."
> > >
> > > The reviewer asks: "Have the authors sent it to the original authors of min-p? What are the responses?"
> > >
> > > Yes. We have already engaged in extensive dialogue with the original authors.
> > >
> > > - We informed them of the omitted baseline data; they silently added it after peer review had concluded.
> > >
> > > - We challenged the insights they drew from their human evaluations; they ran a new one (which we also analyzed) and again drew incorrect conclusions.
> > >
> > > - We questioned their 1.1M GitHub stars & 50k repos claim; they subsequently retracted both claims.
> > >
> > > The original authors have already responded to our findings by altering their paper, yet the fundamental lack of performance improvement remains. A "Right of Reply" does not imply the right to stop valid scientific criticism from being published.
> > >
> > > ## Conclusion
> > >
> > > We have adhered to the highest standards of scientific ethics: we communicated with the original authors, we rigorously tested their method using a new framework (Best-of-N), and we are submitting our findings to the same venue that published the original work. We believe this paper is exactly the kind of "self-correction" mechanism that high-quality science requires.

---

> > > > ### Comment · Reviewer_2vSc · 2025-11-25
> > > >
> > > > I understand the frustration of having wasted time and effort on something, and the desire to publish the lesson or correct the record. But I am not sure we are taking an appropriate approach here (under the current accepted norms of ICLR). We can agree to disagree. I urge the authors to send the draft to the authors of min-p. You do not need their blessings but it's the right thing to ask for their explicit feedback.
> > > >
> > > > I've said enough and I'll let other reviewers and AC chime in from now on.

---

> > ### Comment · Reviewer_2vSc · 2025-11-24
> > **Re: Response to Weakness #3**
> >
> > I understand one can never be 100% certain about most things, not even with mathematical proofs. However, we must take into account the potential damage that our mistake may cause. For a usual ICLR paper, the damage will primarily be on the authors themselves for having published something wrong. For this submission, the potential damage will primarily be on someone else. This is why we must exercise a higher level of care.
> >
> > I strongly believe the original authors must be given a chance to respond to this draft. Have the authors sent it to the original authors of min-p? What are the responses?

---

> ### Author Response · Authors · 2025-11-27
> **Factual Clarification to Reviewer 2vSc**
>
> # Factual Correction: We Have Corresponded Multiple Times with the Authors of Min-P
>
> > I urge the authors to send the draft to the authors of min-p. You do not need their blessings but it's the right thing to ask for their explicit feedback.
>
> To expand on what we told you previously, we have corresponded with the authors of min-p back and forth, privately and publicly in writing multiple times, and have met with them live twice. The authors have updated their manuscript in response to our feedback multiple times. We spent several months integrating their feedback and running additional experiments for our manuscript that we again shared with them.
>
> Please stop telling us to talk to the authors when we had done so repeatedly, and please revise your position.

---

> > ### Comment · Reviewer_2vSc · 2025-11-27
> >
> > I was not planning to reply further, but in case I have accidentally hurt any feelings, let me explain one thing. From your reply
> >
> > > Yes. We have already engaged in extensive dialogue with the original authors. We informed them of the omitted baseline data; they silently added it after peer review had concluded. We challenged the insights they drew from their human evaluations; they ran a new one (which we also analyzed) and again drew incorrect conclusions. We questioned their 1.1M GitHub stars & 50k repos claim; they subsequently retracted both claims. The original authors have already responded to our findings by altering their paper, yet the fundamental lack of performance improvement remains. A "Right of Reply" does not imply the right to stop valid scientific criticism from being published.
> >
> > it was not clear to me if you have shared the draft to the authors of min-p, which is why I made the suggestion (again). If you had earlier said
> >
> > > for our manuscript that we again shared with them
> >
> > I'd have stopped mentioning it. That's all. Not an obsessed freak insisting on telling you what to do.
> >
> > If I am the only outlying reviewer, happy to be convinced otherwise. For now, I'll wait to hear others' opinions.

---

### Official Review · Reviewer_buMc · 2025-10-29

**Soundness:** 2
**Presentation:** 3
**Contribution:** 1
**Rating:** 4
**Confidence:** 3

**Summary:**

This paper presents a detailed case study of min-p, which is a high-visibility paper published in ICLR 2025. The authors have conducted a lot of rigorous experiments and found the results claimed from the min-p no longer hold. The authors thus derive a blueprint for more rigorous empirical ML research. This paper very much reminds me of one of the ML reproducibility challenge papers.

**Strengths:**

- This paper critically examines the validity of empirical machine learning research through carefully designed statistical tests and rigorous experimental design. It offers valuable insights for the community and serves as a warning about the noise and inconsistency in current ML research reporting.

- The paper is clearly written and presented very nicely.

**Weaknesses:**

- This paper could have a greater impact if it positioned itself as a position paper outlining guidelines for standard practices in rigorous ML experimentation. At present, its conclusions are drawn from a single case study, and their external validity depends on whether Min-P is truly representative of broader systemic issues.

**Questions:**

- Regarding 3.1, I get that when controlling for the hyper-parameter tuning budget, there is no distinction between min-p and other sampling methods. Did you run experiments where all the samplers are fully tuned? Does min-p have an advantage in this case?

- The authors are questioning the community adoption. While the original paper may have exaggerated, min-p seems to be well accepted by the community as I can see. Is there further community evidence that does not support min-p’s claim of superiority?

---

> ### Author Response · Authors · 2025-11-20
> **Response to Reviewer buMc**
>
> We thank Reviewer buMc for their positive assessment of our experimental rigor and clear presentation, and for recognizing that this work serves as a valuable warning regarding inconsistency in ML research. We address the specific feedback below.
>
> # Response to Weakness #1: This should be a position paper
>
> The reviewer suggests this work might fit better as a position paper. We offer two counter-arguments:
>
> 1. A Case Study is Essential Proof: Abstract guidelines (common in position papers) are often ignored. By contrast, a rigorous empirical case study of a high-profile ICLR Oral paper forces the community to confront the issue. It demonstrates exactly how standard metrics are being misused and provides a concrete, proven alternative methodology (our "Best-of-N" framework) to fix it.
>
> 2. Incompatible with Position Tracks: This manuscript includes substantial original research (e.g., a new human evaluation study and 6,000 A100-hours of hyperparameter sweeps). This empirical volume actually disqualifies it from standard Position Paper tracks. For example, the ICML/NeurIPS Position Paper Calls for Papers explicitly state:
>
> “Submissions to the main ICML/NeurIPS conference track emphasize original research and novel results. In contrast, submissions to the position paper track will be judged primarily on whether they present a compelling position…”
>
> We selected min-p not just because it is a popular method, but also because it was an ICLR Oral (and the 18th highest ranked submission). This demonstrates that the "systemic issues" the reviewer mentions are present at the highest levels of our field. Using this specific, high-profile example allows us to demonstrate exactly how standard metrics (like win-rates and hyperparameter tuning) are being misused, and—crucially—to provide a concrete alternative methodology (our "Best-of-N" framework) to fix it.
>
>
> # Answer to Question 1: Did you run experiments where all samplers are fully tuned?
>
> **Yes, we did, and no, min-p does not have an advantage.**
>
> This is the central result of Figure 5. In that analysis, we calculated the difference between the best possible min-p score and the best possible baseline score as the tuning budget $N$ increased from 1 to 100.
>
> As $N$ increases (moving to the right on the x-axis), the curves generally converge to zero or slightly below zero. This indicates that once both methods are "fully tuned" (within a budget of 100 distinct configurations), min-p is statistically indistinguishable from—or slightly worse than—standard samplers.
>
> The perceived advantage of min-p in prior work came from comparing a tuned min-p against an untuned or poorly tuned baseline. When both are tuned equally, the advantage vanishes.
>
> # Answer to Question 2: Community Adoption
>
> > min-p seems to be well accepted by the community as I can see
>
> - The community adoption was driven by the strong claims in the original ICLR paper, specifically that min-p was used by 1.1 million stars-worth of repos. These claims were false and retracted by the authors after we challenged them. The "adoption" signal the reviewer sees is largely a lagging indicator of this initial hype.
>
> > Is there further community evidence that does not support min-p’s claim of superiority?
>
> - The reviewer asks why there isn't more community evidence against min-p. In the current research climate, negative results are rarely published. Practitioners who tried min-p and found no benefit likely simply reverted to top-p without writing a paper about it. Our work breaks this cycle by providing the rigorous scientific evidence necessary to correct the record.
>
> # Conclusion
>
> We believe that correcting the scientific record regarding a high-profile ICLR Oral paper is a vital contribution that belongs at ICLR. We hope the clarification regarding the "fully tuned" results (Figure 5) addresses the reviewer's technical concerns.

---

> > ### Comment · Reviewer_buMc · 2025-11-27
> >
> > Thanks for the response.
> >
> > It seems that all reviewers agree that the paper can be reformulated into a position or workshop paper.  Maybe the authors should rather focus on "how standard metrics (like win-rates and hyperparameter tuning) are being misused", instead of only attacking this one single min-p paper.
> >
> > I understand that negative results are seldom reported, but my question is really about whether there’s any evidence from community discussions—forums, user groups, or similar sources. For a method that’s so widely used, if it truly performed poorly, you would expect to see complaints. Many papers contain flaws, and some issues are unintentional and don’t necessarily affect a method’s overall adoption.

---

### Official Review · Reviewer_TzxU · 2025-11-01

**Soundness:** 2
**Presentation:** 2
**Contribution:** 2
**Rating:** 2
**Confidence:** 4

**Summary:**

This paper re-analyzes the ICLR 2025 paper _“Turning Up the Heat: Min-P Sampling for Creative and Coherent LLM Outputs”_, arguing that its main claims about min-p sampling’s superiority are unsupported. The authors identify overstatements in the original work’s human evaluations, benchmark tests, and reporting practices. Using this case study, they propose a blueprint for improving rigor and transparency in empirical ML research.

**Strengths:**

- This paper conducts a detailed and transparent re-examination of a prior work, carefully identifying exaggerations and verifying claims through rigorous statistical and experimental checks.
- The discussion section provides thoughtful and necessary guidance for improving rigor and transparency in scientific research, offering lessons that are broadly applicable across scientific disciplines.

**Weaknesses:**

- Although the authors claim “From this case study, we derive a blueprint for more rigorous research,” the proposed blueprint only appears briefly in the final discussion section (less than one page). This portion is disproportionately small relative to the claimed contribution and lacks the depth or generalization needed to stand as a substantive methodological advance.

- While the manuscript provides an important and well-executed re-analysis of a previous study, it primarily functions as a reproducibility commentary rather than a novel empirical or theoretical contribution. I recognize its educational and community value, but it does not clearly fit within any of the main ICLR paper categories. The authors might consider submitting it as a position paper or workshop contribution, where its insights on research rigor and transparency would be more appropriately framed.

**Questions:**

Please see the weakness.

---

> ### Author Response · Authors · 2025-11-20
> **Response to Reviewer TzxU**
>
> We thank Reviewer TzxU for their detailed assessment, acknowledging our work as a "detailed and transparent re-examination" and noting that our discussion offers "lessons that are broadly applicable." We appreciate the feedback regarding the framing of the "blueprint" and the paper’s category. We address these two main concerns below.
>
> # Response to Weakness #1: The Blueprint Appears Only Briefly in Discussion
>
> The reviewer notes that the textual description of the blueprint in the Discussion is brief. We agree that the summary is concise, but we wish to clarify that the blueprint is the methodology itself, which is detailed throughout Sections 2, 3, and 4. We will make this more clear in the revised manuscript.
>
> Our paper does not merely point out errors; it introduces specific, reusable methodologies for rigorous evaluation and demonstrates how to use them:
>
> - Controlling for Optimization Volume (Section 3 & Fig 4/5): We introduce a formal "Best-of-N" methodology to compare samplers. Most sampling papers fail to control for the volume of hyperparameter search, leading to illusory gains. Our proposed method (equalizing the search space size $N$ and plotting performance vs. $N$) provides a generalizable tool for the community to fairly compare any hyperparameter-dependent algorithm.
>
> - Statistical Rigor in Human Eval (Section 2): We demonstrate a concrete statistical framework (Intersection-Union Tests, Bonferroni corrections) necessary for claiming "consistent" superiority, which is often overlooked in ML human evaluations.
>
> In the final version, we will expand the Discussion section to explicitly link the summary points back to the specific methodologies in Sections 3 and 4, making the "blueprint" a more standalone guide for future authors.
>
> # Response to Weakness #2: Poor Venue Fit
>
> The reviewer suggests this may be better suited as a workshop or position paper. We respectfully argue that this work fits squarely within ICLR’s mission for the following reasons:
>
> 1. Correction of the Scientific Record: The paper we analyze (min-p, Nguyen et al., 2024) was an ICLR Oral presentation. It made strong claims that have influenced the community (evidenced by the initial claim of 1.1M stars and adoption). **If a top-tier venue publishes a flawed method as a top paper, it is the responsibility of that same venue to publish the rigorous correction**. Relegating the correction to a workshop limits its visibility and allows the misconception to persist.
>
> 2. Scientific Progress via Falsification: As noted in our introduction (referencing Schaeffer et al., 2025; Pineau et al., 2017), the field faces a replication crisis. "Novelty" at ICLR should not be limited to proposing new algorithms; it must encompass empirical rigor that invalidates ineffective methods, especially methods that ICLR itself highlighted. Establishing that a popular new method offers no advantage over baselines is a significant empirical contribution that saves community resources.
>
> 3. Precedent: There is a strong tradition of "critique and re-analysis" papers at premier venues. For example, Agarwal et al. (2021) on RL evaluation at NeurIPS, Carlini et al. (2018) on obfuscated gradients in adversarial robustness at ICML, and Haibe-Kains et al. (2020) on AI for medicine in Nature. Similar to these works, our paper goes beyond simple "reproducibility" (running the code) to perform "replicability and validity" analysis, showing the conclusions do not follow from the data and providing practical guidance for others.
>
> # Conclusion
>
> We believe this paper offers dual value: immediate correction of a high-visibility error in LLM sampling, and a long-term methodological contribution via the "Best-of-N" evaluation framework. We hope the reviewer agrees that maintaining the rigor of ICLR publications is a task best suited for ICLR itself.

---

### Author Response · Authors · 2025-12-04
**Concluding Remarks to AC: Why ICLR Must Correct the Scientific Record**

We thank the reviewers for their engagement and for acknowledging that our work is detailed, transparent, and a “comprehensive re-examination” that exposes a “crisis of rigor” in the field.

While the reviewers unanimously agree that our analysis is technically sound and that the conclusions of the original min-p paper are invalid, they have expressed hesitation regarding (1) venue fit (suggesting a workshop or position paper) and (2) fairness (concerns about criticizing a single paper).

We respectfully argue that rejecting this paper on these grounds would set a dangerous precedent for scientific integrity at ICLR. We ask you to consider the following three points in your decision:

# 1. Institutional Responsibility

The paper we analyze (min-p, Nguyen et al., 2024) was not an obscure submission; it was an ICLR 2025 Oral presentation, and the 18th highest scoring submission. It made bold claims of superiority based on omitted baseline data, improper statistics, and community adoption figures (1.1M stars) that were subsequently retracted. If ICLR grants a high-profile platform to a flawed method, ICLR bears the responsibility to host the rigorous correction. Relegating this correction to a workshop or a different venue effectively buries the scientific truth and allows the error to persist in the top-tier record.

# 2. Novel Methodological Contribution (Beyond "Just a Critique")

Reviewers suggested this is merely a reproduction study. This is incorrect. Beyond invalidating min-p, we introduce a generalizable methodology: the "Best-of-N" evaluation framework (Section 3, Figs 4 & 5). This framework solves a pervasive problem in the community—unfair comparisons due to unequal hyperparameter tuning volumes. This offers concrete, forward-looking value to the community, aligned with prior influential critique papers at top venues (e.g., Engstrom et al., ICLR 2020; Carlini et al., ICML 2018).

# 3. Ethical Rigor and Fairness

Reviewer 2vSc expressed concern that the original authors were not given a "right of reply." As detailed in our rebuttals, we engaged in extensive dialogue with the original authors prior to submission. They have already reacted to our findings by adding previously omitted data and retracting false claims in their camera-ready version. We have adhered to the highest standards of professional ethics; a "right of reply" does not equate to immunity from valid scientific refutation.

# Conclusion

The community is facing a replication crisis. This paper does exactly what we say we want to see in ML research: it checks the work, identifies the flaw, proposes a better evaluation standard (Best-of-N), and corrects the record. We urge the Area Chair to accept this paper to demonstrate that ICLR values rigorous science over hype.

---

### Meta-Review · Area_Chair_yngx · 2026-01-07

**Summary:**

After carefully checking the paper, the reviews, the rebuttal, and the author-reviewer discussions, I think the weak points outweight the strong points. All reviewers pointed out concerns regarding the novelty of this work, and the distinction from prior work is not sufficiently clear. Therefore, I recommend that this paper be considered with caution. The weaknesses are not likely to be fixed in the camera-ready version. Thus, I recommend rejecting this paper.

**Reviewer Concerns:**

The score remains unchanged. I think author didn't address the novelty problem. I have carefully read the rebuttal. The rebuttal does not address any important concern raised by the reviewers.

**Reviewer Scores:**

The score remains unchanged. I have carefully read the rebuttal. The rebuttal does not address any important concern raised by the reviewers.

---

### Decision · Program_Chairs · 2026-01-26

Reject